# Impairment of Endogenous Synthesis of Omega-3 DHA Exacerbates T-Cell Inflammatory Responses

**DOI:** 10.3390/ijms24043717

**Published:** 2023-02-13

**Authors:** Emanuela Talamonti, Anders Jacobsson, Valerio Chiurchiù

**Affiliations:** 1Department of Biochemistry and Biophysics, Stockholm University, 114 Stockholm, Sweden; 2Department of Molecular Biosciences, The Wenner-Gren Institute, Stockholm University, 114 Stockholm, Sweden; 3Institute of Translational Pharmacology, National Research Council, 00133 Rome, Italy; 4Laboratory of Resolution of Neuroinflammation, IRCCS Santa Lucia Foundation, 00179 Rome, Italy

**Keywords:** fatty acids, omega-3, DHA, elongase, T lymphocytes, dendritic cells

## Abstract

Omega-3 (ω-3) polyunsaturated fatty acids, including docosahexaenoic acid (DHA), are involved in numerous biological processes and have a range of health benefits. DHA is obtained through the action of elongases (ELOVLs) and desaturases, among which Elovl2 is the key enzyme involved in its synthesis, and can be further metabolized into several mediators that regulate the resolution of inflammation. Our group has recently reported that ELOVL2 deficient mice (Elovl2^−/−^) not only display reduced DHA levels in several tissues, but they also have higher pro-inflammatory responses in the brain, including the activation of innate immune cells such as macrophages. However, whether impaired synthesis of DHA affects cells of adaptive immunity, i.e., T lymphocytes, is unexplored. Here we show that Elovl2^−/−^ mice have significantly higher lymphocytes in peripheral blood and that both CD8+ and CD4+ T cell subsets produce greater amounts of pro-inflammatory cytokines in both blood and spleen compared to wild type mice, with a higher percentage of cytotoxic CD8+ T cells (CTLs) as well as IFN-γ-producing Th1 and IL-17-producing Th17 CD4+ cells. Furthermore, we also found that DHA deficiency impacts the cross-talk between dendritic cells (DC) and T cells, inasmuch as mature DCs of Elovl2^−/−^ mice bear higher expression of activation markers (CD80, CD86 and MHC-II) and enhance the polarization of Th1 and Th17 cells. Reintroducing DHA back into the diets of Elovl2^−/−^ mice reversed the exacerbated immune responses observed in T cells. Hence, impairment of endogenous synthesis of DHA exacerbates T cell inflammatory responses, accounting for an important role of DHA in regulating adaptive immunity and in potentially counteracting T-cell-mediated chronic inflammation or autoimmunity.

## 1. Introduction

Docosahexaenoic acid (DHA) is a ω-3 polyunsaturated fatty acid (PUFA), which is not only a key structural component of membrane phospholipids but also plays several biological functions, making it one of the most important bioactive lipids [1,2]. DHA is synthesized from the dietary essential fatty acid α-linolenic acid after several elongation and desaturation reactions performed by distinct enzymes residing in the endoplasmic reticulum [3]. The key enzyme in DHA biosynthesis is elongation of long-chain fatty acids 2 (ELOVL2), which is mainly expressed in the liver, testis, retina, and central nervous system (CNS), all of which are tissues particularly rich in DHA [3,4]. DHA is essential for CNS functions, including neuronal survival [5], synaptic functions, and neurotransmission [6], as well as protecting from neuroinflammation and cognitive decline [7]. Indeed, a DHA-deficient diet is involved in the pathophysiology of memory impairment [8] and several mood disorders, such as anxiety and depressive-like behaviors [9,10].

Furthermore, increasing evidence and longitudinal prospective cohort studies have recently demonstrated the role of DHA in controlling inflammatory responses [11], also after the discovery that it can be metabolically converted into many anti-inflammatory and pro-resolving mediators such as resolvins, protectins, and maresins [12] and that this can impact the functions of both myeloid cells such as macrophages and glial cells [13,14] but also of T cells [15]. Indeed, we have recently shown that mice with impaired systemic DHA synthesis (Elovl2^−/−^) were not only characterized by an increased expression of pro-inflammatory cytokines in the brain [16], but also showed an altered M1/M2 macrophage phenotype, with M1 being more pro-inflammatory and M2 being less protective [17]. Furthermore, more recent evidence showed an effect of DHA on modulating T lymphocytes either directly or through an indirect control of dendritic cells [18,19,20]. However, the impact of a deficiency in systemically biosynthesizing DHA on T cells is still unknown. Hence, here we sought to systematically investigate whether Elovl2^−/−^ mice, characterized by a systemic impairment of endogenous DHA synthesis, displayed altered T-cell responses and whether such alterations could be restored upon DHA supplementation.

## 2. Results

### 2.1. DHA Deficiency Increases Lymphocyte Cell Count

Since blood DHA levels are significantly reduced in Elovl2^−/−^ mice, as shown in our previous findings [4,21], and given the overgrowing evidence of the role of DHA in controlling inflammatory responses, we initially examined several blood parameters, including both red and white blood cells.

A complete blood count revealed that hemoglobin, hematocrit and other specific corpuscular indices such as MCV and MCHC remained unchanged between WT and Elovl2^−/−^ mice (Appendix A).

Furthermore, when analyzing the white blood cell differential, Elovl2^−/−^ mice showed a slight increase in the total number of leukocytes (Figure 1a), and their distribution into the different cell subsets revealed that only the lymphocyte population was significantly increased (Figure 1e). Basophils (Figure 1d) and monocytes (Figure 1f) showed decreased or increased trends, that were not yet significant, whereas other cell populations, namely neutrophils and eosinophils (Figure 1b,c) showed no variation between WT and Elovl2^−/−^ mice (Figure 1).

### 2.2. DHA Deficiency Exacerbates the Pro-Inflammatory Responses of T Lymphocytes In Vitro

Given that lymphocytes were the only peripheral blood cell population that was significantly higher in Elovl2^−/−^ mice, and that over 90% of lymphocytes are T cells, we next questioned whether this transgenic mouse model displayed a more inflammatory T cell immunophenotype in the absence of DHA in their diet. To do so, we set up two different experimental protocols for T cell activation, by stimulating T cells isolated from peripheral blood and spleen with either PMA+Ionomycin (Appendix A) or with the more physiological polyclonal anti-CD3 and anti-CD28 (Appendix A and Figure 2a). As shown in Appendix A and as expected, upon PMA+Ionomycin stimulation both CD8+ and CD4+ T cell subsets displayed a prominent increase in the production of pro-inflammatory cytokines than unstimulated cells from both blood and spleen. In particular, such increased production of cytokines was specifically observed in the main T cell subsets, namely cytotoxic CD8+ T cells (CTLs) that produce TNF-α and IFN-γ and already committed IFN-γ-producing T-helper (Th)1 and IL-17-producing Th17 CD4+ T cells. Interestingly, CTLs, Th1 and Th17 from Elovl2^−/−^ mice consistently produced significantly higher amounts of their respective and signature cytokines (Appendix A). No sex-specific difference was observed, since T-cell responses in both male and female mice were comparable.

### 2.3. DHA Deficiency Exacerbates the Pro-Inflammatory Responses of T Lymphocytes In Vivo

A similar scenario was observed upon polyclonal stimulation with the more physiological anti-CD3 and anti-CD28 polyclonal stimuli, where all T-lymphocytes subsets showed a significant increase in TNF and IFN-γ from CTLs as well as in IFN-γ from Th1 and IL-17 from Th17 T cells as observed in the different heatmaps (Figure 2a,b). Of note, when DHA was supplemented in the diet for 2 months, the levels of the signature cytokines produced by CTLs, Th1 and Th17, obtained from both blood and spleen, were significantly reduced, and returned to levels that were comparable to those of WT control mice, independently of the type of stimulation (Appendix A and Figure 2a,b). Since these findings were obtained from in vitro stimulation of peripheral blood T lymphocytes, we next sought to ascertain if this evidence could also be replicated in vivo by intraperitoneally injecting mice with anti-CD3 (50 µg per mouse) and collecting blood after 3 h (Figure 3a). We found that CTLs (CD8+) (Figure 3b) and Th1(CD4+) cells produced higher amounts of TNF and IFN-γ (Figure 3c), while Th17 cells produced more IL-17 (Figure 3c) in Elovl2^−/−^ mice compared to WT control mice, and this was reversed in mice supplemented with DHA (Figure 3a–c), suggesting that in absence of endogenously produced DHA, the pro-inflammatory responses of all T cell subsets are exacerbated.

### 2.4. Impacts of DHA Deficiency on DC-T Cell Cross-Talks

Since the activity of T cells is mainly driven by antigen-presenting cells (APCs), among which dendritic cells are the most potent APCs, we next sought to investigate whether DHA deficiency also had an impact on this immune synapse. At first, we examined the maturation level of dendritic cells of Elovl2^−/−^ mice compared to WT control mice in absence of DHA in their diet. To do so, we set up a differentiation and maturation protocol in which monocytes were differentiated into immature DC (immDC) and then activated into mature DC (matDC), and we found that only matDC presented significantly higher levels of their activation markers CD80, CD86, and MHC-II (Figure 4b), while comparable levels of these markers were observed in immDC cells (Figure 4a). Notably, upon setting up a co-culture between matMo-DC and allogenic enriched T cells and subsequently analyzing T cells by flow cytometry, we found that T cells that were in contact with matMo-DC derived from Elovl2^−/−^ mice showed higher expression levels of inflammatory cytokines from TNF/IFN-γ-producing CTLs, IFN-γ-producing Th1 and IL-17-producing Th17 cells (Figure 4c). Interestingly, when co-cultivating matMo-DC obtained from Elovl2^−/−^ mice underwent DHA supplementation in their diet with T cells, the proinflammatory cytokine responses of the latter were significantly reduced for IFN-γ-producing Th1 and IL-17-producing Th17 cells (Figure 4c), even though TNF production from both of these T-helper subsets was not affected, nor was TNF and IFN-γ production from CTLS. These data suggest that DHA is partly required for the modulation of DC-mediated T cell responses, specifically T helper cell responses.

## 3. Discussion

The evidence that incorporating omega-3 fatty acids into our daily diet is of great importance for the prevention and treatment of a number of human diseases, such as cardiovascular and neurological disorders, also by having anti-inflammatory properties, is undisputed. Among them, DHA bears the most beneficial effects, especially in the brain, heart, and immune system [22,23,24,25], even if its biosynthesis seems to be exclusively operated by specific organs and tissues, mainly the liver, and subsequently distributed to the other organs, such as immune cells, that are unable to synthesize DHA due to a lack of Elovl2 expression. We have recently discovered that genetic impairment of Elovl2-mediated DHA synthesis impacts macrophage plasticity and polarization by sustaining the hyperactivation of M1-like macrophages and promoting a M2-to-M1-like immunophenotype switch [17]. However, the specific role of DHA on the activation status and the immunological responses of adaptive immunity, and particularly T lymphocytes, is still scarce. Hence, in this study, we interrogated whether endogenously synthesized DHA could affect T cell responses. To do this, we used mice deficient for Elovl2, and we analyzed the main subsets of T cells, namely not only CD8 and CD4 cells but also T helper subsets, i.e., Th1 and Th17. Notably, we observed that in absence of DHA T cells, not only was the immune cell population significantly altered in terms of cell count, but their inflammatory responses were also impacted. Indeed, although both WT and Elovl2^−/−^ mice were able to efficiently produce all the pro-inflammatory cytokines of the blood and splenic CD8+ and CD4+ subsets, including the signature Th1 and Th17 ones, their levels were significantly higher in Elovl2^−/−^ mice regardless of the type of stimulus, namely general stimulation with PMA/Ionomycin or polyclonal stimulation with anti-CD3 and anti-CD28, suggesting that DHA deficiency makes T lymphocytes more responsive to activatory stimuli in vitro. The same cytokine profiles were also corroborated in vivo upon injection with T-cell activator anti-CD3. The observed exacerbated proinflammatory response of T-cells in Elovl2^−/−^ mice is of pivotal importance because it suggests a role for DHA not only in keeping these adaptive immune cells activated within their boundaries but also in avoiding potential chronic inflammation and autoimmunity. Indeed, Cytotoxic CD8+ T cells (CTLs) eliminate neoplastic, infected, or damaged cells mainly through the release of cytotoxins and potentiate innate immune responses [by macrophages and natural killer (NK) cells] through the release of cytokines such as TNF and IFN-γ [26]. Our data suggests that DHA might play a direct role in blunting the CTL responses during acute inflammation but also indirectly by avoiding further recruitment or activation of innate cells, thus avoiding the onset of chronic inflammation or immune-mediated damage. Furthermore, CD8+ cells can also prime naïve and restimulate experienced CD4+ T cells to release high levels of helper cytokines [27]. Th cells develop from naïve CD4+ T cells and differentiate into specialized subsets after encountering foreign or autoantigens [28,29]. However, persistent or uncontrolled Th cell responses are often associated with pathological states and tissue damage. In particular, excessive and/or abnormal Th1 and Th17 cell responses are involved in chronic inflammation and mediate several autoimmune diseases, including multiple sclerosis, rheumatoid arthritis, and psoriasis [30,31]. Of note, decreased omega-3 PUFAs are associated with such autoimmune diseases, especially multiple sclerosis, where Th1/Th17 cells play a key role in the pathogenesis [32,33,34]. The fact that DCs are able to enhance Th1 and Th17 cells and DHA plays a critical role in this suggests that these antigen presenting cells might act on the ability of this omega-3 PUFA to skew the polarization of naïve T cells into Th cells and this is worth further mechanistic characterization. Our present results suggest that DHA can not only directly modulate the inflammatory responses of committed Th1 and Th17 cells but also can reduce their generation indirectly through the interaction of dendritic cells, which are the most potent antigen-presenting cells. This hypothesis was confirmed by reinstating DHA in the diet and observing that CTLs, Th1, and Th17 responses returned to normal levels as in WT mice. A limitation of this study is represented by the lack of information on the other main subset of CD4 T cells, i.e., regulatory T cells. Our observed modulation of DHA in exacerbated Th1/Th17 responses might also be due to a potential effect on enhancing Tregs. These findings are in line with several recent evidence on the role of omega-3 PUFAs like DHA and EPA in modulating T cell functions in vitro or in vivo [35,36,37] as well as following dietary supplementation [20,38,39,40]. Although we cannot define the mechanism by which DHA deficiency impacts T cell responses in this study, it is plausible that the purported DHA receptors GPR40, GPR120 and PPAR-γ are involved, given that these receptors have recently been associated to Th functions [41,42,43,44]. Notably, since DHA can be quickly metabolized into the specialized pro-resolving mediators (i.e., resolvins, protectins, and maresins), whose role has been recently reported by our group to potently blunt CTLs and Th1 and Th17 cells [15,45,46], this might be an additional possible mechanism for our observed effects in DHA-deficient mice.

## 4. Conclusions

In conclusion, this study highlights the importance of endogenous DHA in the direct and indirect modulation of T cell responses both in vitro and in vivo, suggesting that the enzymatic machinery that leads to DHA biosynthesis is important for the maintenance of adaptive immune functions in order to avoid subsequent chronic inflammation or autoimmunity.

## 5. Materials and Methods

### 5.1. Animals and Tissue Processing

Elovl2^−/−^ mice were generated as described previously [47] and backcrossed into 129S2/Sv for five generations. All animals (male and female) were housed at room temperature and maintained on a 12 h light/dark cycle. Mice that were 20–25 weeks old were fed either standard chow DHA-free diet (10% kcal fat, D12450H, Research Diets, New Brunswick, NJ, USA) or a DHA-enriched diet (10% kcal fat, 1% DHA, D13021002, Research Diets, New Brunswick, NJ, USA) for 3 months, according to the experimental groups. Dietary fatty acid composition is shown in Table 1. All animals were fed ad libitum and had free access to water. At the end of the study, animals were euthanized with CO_2_ and sacrificed by cervical dislocation. Spleen, bone marrow, and peripheral blood were collected immediately after for further processing and flow cytometry studies or cell cultures. Studies were carried out with ethical permission from the Animal Ethics Committee of the North Stockholm region, Sweden, and performed in accordance with national guidelines and regulations for the care and use of laboratory animals in agreement with the guidelines of the European Communities Council Directive 2010/63/EU for the care and use of laboratory animals.

### 5.2. In Vitro and In Vivo Stimulation of T Cells

Peripheral blood mononuclear cells (PBMCs) were isolated from spleen and peripheral blood of wild-type (WT), Elovl2^−/−^ mice and Elovl2^−/−^ + DHA mice. In order to measure the intracellular cytokine levels, secretion was inhibited by adding 1 µg/mL brefeldin A (Sigma-Aldrich, Burlington, MA, USA) 5 h before the end of stimulation with either PMA/Ionomycin or Dynabeads CD3/CD28 T Cell Expander (one bead per cell; Invitrogen, Waltham MA, USA) [48,49]. For in vivo stimulation, mice were intraperitoneally (i.p.) injected with 50 µg of anti-CD3 (Armenian hamster IgG, clone 145-2C11) (Biolegend, San Diego, CA, USA). Blood samples were recovered 3 h after antibody injection, and a cell suspension was prepared for flow cytometry analysis [48,49,50,51].

### 5.3. Isolation and Culture of Bone Marrow Dendritic Cells

Bone marrow-derived dendritic cells were prepared by culturing bone marrow cells obtained from the femur and tibia of mice in complete RPMI 1640 medium with 5% FBS. Immature dendritic cells (immMo-DC) were obtained by adding 10 ng/mL GM-CSF and 20 ng/mL IL-4 for the first 6 days and changing the medium every 2 days. Subsequently, immMo-DC were challenged with 100 ng/mL LPS plus 10 ng/mL IFN-γ for 2 additional days to obtain mature dendritic cells (matMo-DC) [49].

### 5.4. Co-Cultures between Dendritic Cells and T Cells

After negative selection of all antigen presenting cells, mat-Mo-DC derived from Elovl2^−/−^ mice were tested for their ability to activate allogenic naïve T cells (1 × 10^5^/well) obtained from WT mice. Briefly, matMo-DC were cultured with T cells for 24 h in 96-well round-bottom plates at 37 °C at a DC:T-cell ratio of 1:10 in 200 uL final volumes of X-VIVO 15 medium (Lonza, Walkersville, MD, USA), as previously reported [49,50]. After an overnight incubation, 1 µg/mL brefeldin A was added for 5 h, and then cells were collected and analyzed by flow cytometry.

### 5.5. Flow Cytometry

For T cell immunophenotyping, total PBMCs were stained at the cell surface with PE-Cy7-conjugated anti-CD4 (1:100, eBioscience, Waltham, MA, USA), FITC-conjugated anti-CD8 (1:100, Miltenyi Biotec, Bergisch Gladbach, Germany), permeabilized with Cytofix/Cytoperm reagents (BD Biosciences, Franklin Lakes, NJ, USA), and then stained intracellularly with APC-conjugated anti-TNF-α (1:100, eBioscience), v450-conjugated IFN-γ (1:50, eBioscience) and PE-conjugated anti–IL-17 (1:30, eBioscience) in 0.5% saponin at room temperature for 30 min. For each analysis, at least 100,000 live cells were acquired by gating on Pacific Orange–conjugated Live/Dead negative cells, as reported [45,48]. Cells were analyzed using FACSVerse (BD Bioscience, Franklin Lakes, NJ, USA).

### 5.6. Statistical Analysis

The data were presented as mean ± SEM and were analyzed by means of Prism 9 software (GraphPad Software, San Diego, CA, USA). Flow cytometry data was analyzed with FlowJo Software (TreeStar, Ashland, OR, USA). Differences between two or more groups were analyzed by means of a Student’s *t* test or ANOVA followed by a Bonferroni post hoc test, respectively. A *p* value < 0.05 was considered significant.

## Figures and Tables

**Figure 1 ijms-24-03717-f001:**
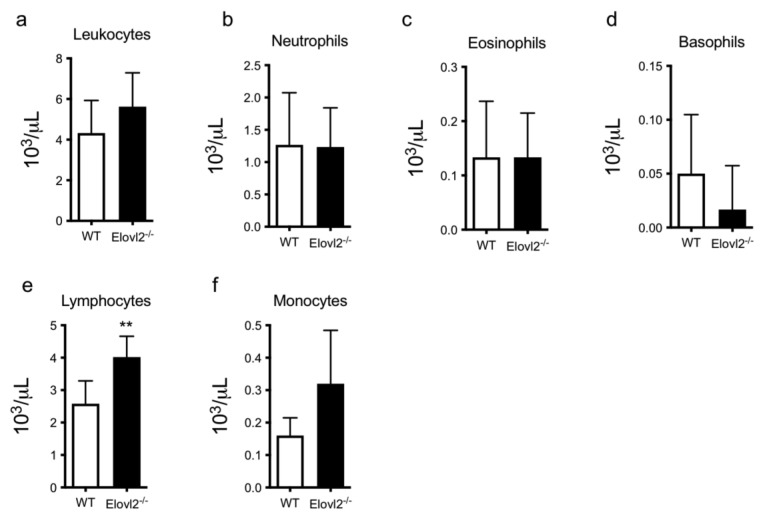
Cell count in peripheral blood of (**a**) total leukocytes, (**b**) neutrophils, (**c**) eosinophils, (**d**) basophils, (**e**) total lymphocytes, and (**f**) monocytes in WT and Elov2^−/−^ mice. ** *p* < 0.01.

**Figure 2 ijms-24-03717-f002:**
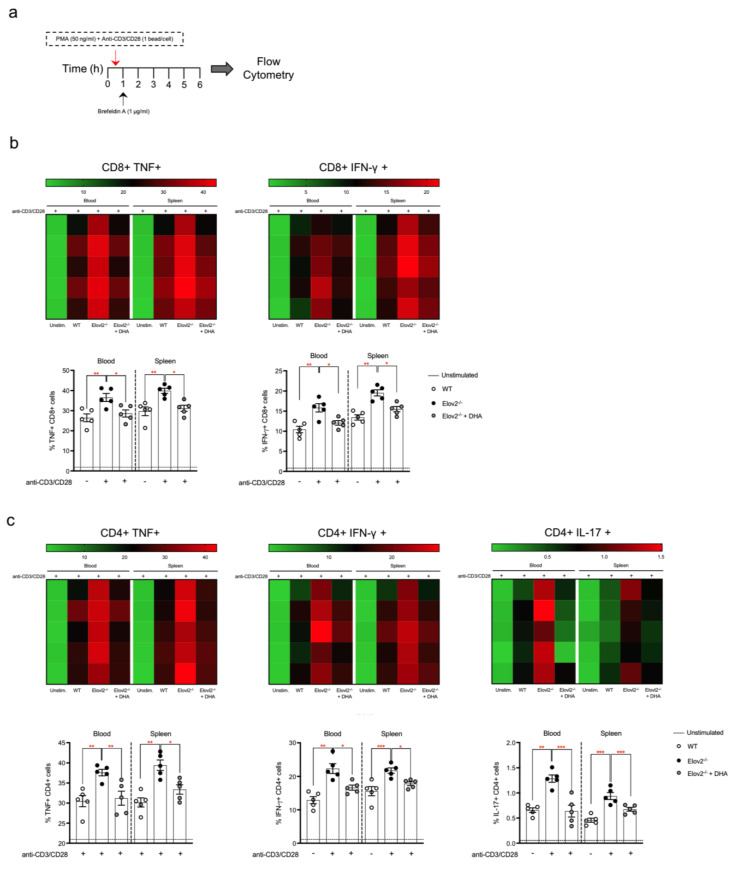
(**a**) Schematic representation of the experimental design in vitro; (**b**) heatmaps and histograms of intracellular TNF-α and IFN-γ production from CD8+ following stimulation with anti-CD3/CD28 beads in the blood and spleen of WT, Elov2^−/−^ and Elov2^−/−^ + DHA. (**c**) heatmaps and histograms of intracellular IFN-γ and IL-17 production from CD4+ following stimulation with anti-CD3/CD28 beads in the blood and spleen of WT, Elov2^−/−^ and Elov2^−/−^ + DHA. Heatmaps show the percentage of intracellular cytokine expression as determined by flow cytometry staining. * *p* < 0.05; ** *p* < 0.01; *** *p* < 0.005.

**Figure 3 ijms-24-03717-f003:**
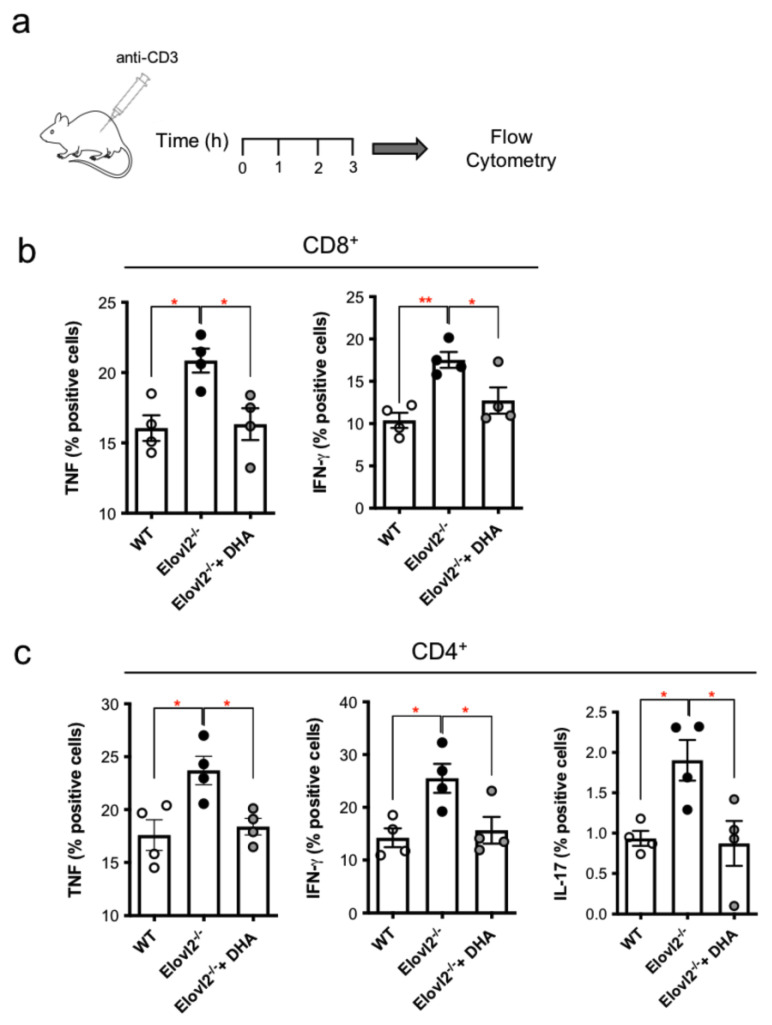
(**a**) Schematic representation of the experimental design in vivo; (**b**) histograms of intracellular TNF-α and IFN-γ production from CD8+ following in vivo injection of anti-CD3 in WT, Elov2^−/−^ and Elov2^−/−^ + DHA. (**c**) histograms of intracellular IFN-γ and IL-17 production from CD4+ following in vivo injection of anti-CD3 in WT, Elov2^−/−^ and Elov2^−/−^ + DHA. * *p* < 0.05; ** *p* < 0.01.

**Figure 4 ijms-24-03717-f004:**
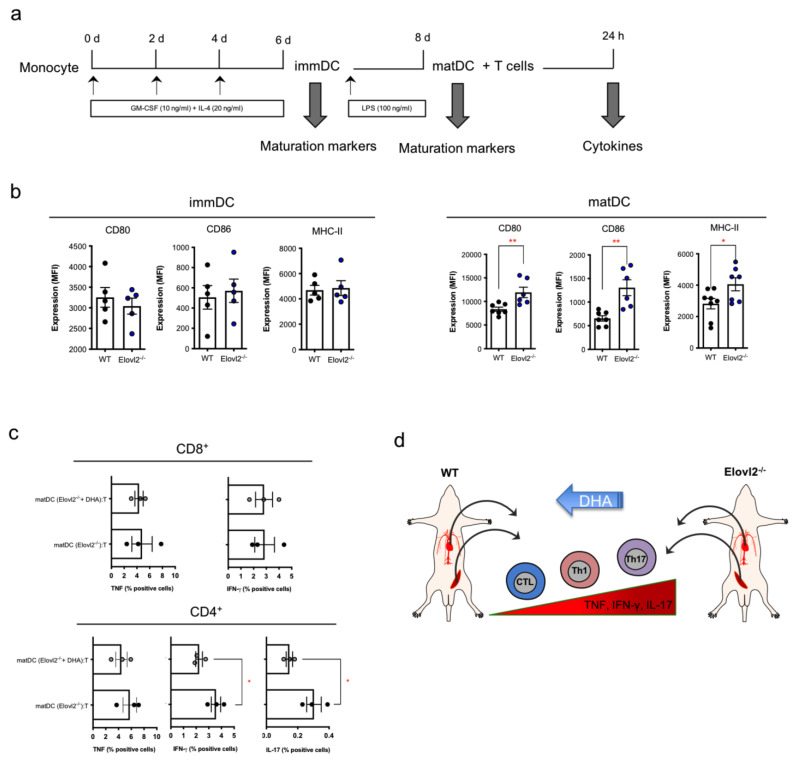
(**a**) Schematic representation of the experimental design; (**b**) histograms of surface expression of CD80, CD86, and MHC-II in immDC and matDC of WT, Elov2^−/−^, and Elov2^−/−^ + DHA. (**c**) histograms of intracellular TNF-α and IFN-γ production from CD8+ and IFN-γ and IL-17 production from CD4+ following co-culture with matDC obtained from Elov2^−/−^ and Elov2^−/−^ + DHA. (**d**) Summary of DHA-induced influence on T-cell subsets. * *p* < 0.05; ** *p* < 0.01.

**Table 1 ijms-24-03717-t001:** Dietary fatty acid composition.

Fatty Acid	Not DHA-Enriched Diet D12450H	DHA-Enriched Diet D13021002
C12:0	0.09	0.05
C14:0	0.80	0.48
C15:0	0.05	0.05
C16:0	16.83	12.48
C16:1n-9	0.17	0.09
C16:1n-7	0.86	0.51
C18:0	8.43	5.35
C18:1n-9	27.87	21.35
C18:1n-7	1.66	1.29
**C18:2n-6** *	**37.43**	**35.64**
C18:3n-6	data	data
**C18:3n-3** **	**4.24**	**4.46**
C18:4n-3	0.05	0
C20:0	0.30	0.23
C20:1n-9	0.37	0.25
C20:2n-6	0.31	0.19
C20:4n-6	0.11	0.24
C20:4n-3	0	0.03
C20:5n-6	0	0.71
C22:0	0.24	0.20
C22:3n-3	0	0.08
C22:4n-6	0	1.04
C22:5n-3	0	0.57
**C22:6n-3**	**0**	**14.53**

* LA (linoleic acid) ** ALA (alpha-linolenic acid) ** DHA (docosahexaenoic acid).

## Data Availability

Not applicable.

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
