# Peer review of "Impairment of Endogenous Synthesis of Omega-3 DHA Exacerbates T-Cell Inflammatory Responses"

_ijms, 2023, doi:10.3390/ijms24043717_

Round 1
Reviewer 1 Report
The manuscript is hypothesis driven and nicely describe the T cell phenotype of the DHA deficient mice. The experiments are well planned and the conclusions are appropriate. The mechanism behind the obtained data is proposed but not proved. 2.1 Basophils are not significantly reduced, to describe it as no variation is a bit superficial. The 2.2 section of results is a complete mixture of the in vitro and in vivo data, which should be separated and explained with the in vitro results followed by the in vivo obtained ones. The legend of Fig 2 does not make clear that it presents in vitro data. The role of DHA via regulating the generation of TH1 and Th17 cells indirectly acting on dendritic cells is a significant observation and worth for further characterization.
Author Response
The manuscript is hypothesis driven and nicely describe the T cell phenotype of the DHA deficient mice. The experiments are well planned and the conclusions are appropriate. The mechanism behind the obtained data is proposed but not proved. 2.1 Basophils are not significantly reduced, to describe it as no variation is a bit superficial. The 2.2 section of results is a complete mixture of the in vitro and in vivo data, which should be separated and explained with the in vitro results followed by the in vivo obtained ones. The legend of Fig 2 does not make clear that it presents in vitro data. The role of DHA via regulating the generation of TH1 and Th17 cells indirectly acting on dendritic cells is a significant observation and worth for further characterization.
R: We thank the Reviewer for the appreciation of our manuscript and for the useful comments.
We modified the sentence on basophils to make the results clearer between all the cell populations.
As suggested, we also separated the in vitro from in vivo results in two different paragraphs (2.2 and 2.3) and we added these terms (in vitro and in vivo) in both figures 2 and 3. We also added a few lines in the discussion on the role of DHA in DC-induced Th1/17 action.
Reviewer 2 Report
Some chronic diseases and weaker immunity function constitute a major problem of public health that is associated with increased risk of mortality and poor quality of life, especially in life-length point of view. In this aspect, this study reported the effects of n-3 PUFAs in the selected white blood cells, especially in T cells. The association among the alterations of immune responses were evaluated in the Elovl2-/- mice model. The study offered some evidence that DHA presents the potential benefits in n-3 PUFA deficiency response, however, I do have some questions raised.
1. First of all, what is aim of this work would be liked to express? It is lack of direct connection of biosynthesis of n-3 LCPUFAs and immunity systems in a specific immune disease. Decreased immune systems have been noted and highly associated in some chronic diseases. Authors should address the immune diseases which caused by the deficiency of n-3 LCPUFA involved mechanism.
2. Is there any observational study or epidemiological evidence to express the relationship between immunity and n-3 LCPUFA? Authors should express more information of these considerations.
3. Why authors presented DHA, but not EPA? The ratio of biosynthesis of n-3 LCPUFA, especially DHA is about 7%, is low in human study. Use Elovl2-/- mice model to reflect all tissues might be able to persuade the DHA deficiency for whole systemic cells.
4. Overall, this was a novel work. Some solid data should be provided and reorganized to focus on the interaction of n-3 PUFAs and involved inflammatory mechanism. I do not recommend it in current status.
Author Response
Some chronic diseases and weaker immunity function constitute a major problem of public health that is associated with increased risk of mortality and poor quality of life, especially in life-length point of view. In this aspect, this study reported the effects of n-3 PUFAs in the selected white blood cells, especially in T cells. The association among the alterations of immune responses were evaluated in the Elovl2-/- mice model. The study offered some evidence that DHA presents the potential benefits in n-3 PUFA deficiency response, however, I do have some questions raised. Overall, this was a novel work. Some solid data should be provided and reorganized to focus on the interaction of n-3 PUFAs and involved inflammatory mechanism. I do not recommend it in current status.
1-2. First of all, what is aim of this work would be liked to express? It is lack of direct connection of biosynthesis of n-3 LCPUFAs and immunity systems in a specific immune disease. Decreased immune systems have been noted and highly associated in some chronic diseases. Authors should address the immune diseases which caused by the deficiency of n-3 LCPUFA involved mechanism. Is there any observational study or epidemiological evidence to express the relationship between immunity and n-3 LCPUFA? Authors should express more information of these considerations.
R: In this study we only wanted to investigate for the first time ever whether a genetic deficiency in synthesizing DHA (not all n-3 LCPUFAs but specifically DHA due to the high specificity of Elovl2) on T cell responses. For decades it is indeed known that n-3 deficiency in the diet is detrimental for all chronic inflammatory diseases, including autoimmune diseases, but not direct evidence was observed on the role of DHA on specific T cell subsets and on the inability to synthesize DHA (from alpha-linolenic acid and EPA). For example, is known from a meta-analysis that n-3 LCPUFAs deficiency is associated to multiple sclerosis (MS), the main autoimmune disease caused by T cells and we also performed lipidomics in the blood of MS patients in our 2020 paper and we found reduced levels of DHA and EPA. All this was added in the discussion.
3-4.Why authors presented DHA, but not EPA? The ratio of biosynthesis of n-3 LCPUFA, especially DHA is about 7%, is low in human study. Use Elovl2-/- mice model to reflect all tissues might be able to persuade the DHA deficiency for whole systemic cells.
R: we presented DHA and not EPA because Elvol2 enzyme is only involved in the biosynthesis of DHA from EPA, so mice with Evolv2 genetic deficiency have normal levels of EPA but cannot convert it into DHA. Also, EPA doesn’t have potent anti-inflammatory effect because it is metabolized into >30 mediators which are all of pro-inflammatory nature (i.e. prostaglandins PGE3, PGD3, leukotriens LTB5 and thromboxanes TXB3) except for the 4 resolvins (RvE1-RvE4). However, DHA is metabolized into >30 anti-inflammatory and pro-resolving mediators (resolvins, protectins, maresins, etc.) and that could be the reason of its beneficial effects, as stated in the manuscript.
Reviewer 3 Report
In this manuscript (IJMS-2195689), Talamonti et al., tried to build upon their previous studies on docohexaenoic acid (DHA) and immune system and investigated T-cell-mediated inflammatory responses in particular. The work is nicely conducted and presented and is of significance to a wide field of investigators. However, few concerns remain and needs to be addressed.
Major concerns:
1. The authors used ELOVL2-/- mice, and analyzed PBMCs and spleen to isolate and characterize T cells. They also need to investigate thymus, to overrule any developmental defects with respect to the T cells.
2. In Figures 2B and 2C, authors provide heatmaps. But, do not mention what they measure RNA or protein? There is no scale either and just visual quantification does not do justice.
3. It will be of great help if authors show in main figures the RNA levels of the main cytokines which they study.
4. The authors studied Th1 and Th17. Is there any effect of DHA on the regulatory T cells? The authors need to show and discuss this.
5. IL-6 is a critical driver of Th17 cells. What happens to IL6 in absence of DHAs?
6. Authors need to show the status (expression or activity) of the critical transcription factors which are known to drive Th1, Th17 production and function.
7. Authors correctly point out that the major tissue of DHA production is the Liver. It will be of immense significance if authors can genetically manipulate ELOVL2 levels solely in liver e.g. adenoviral-based methods and demonstrate whether this affects T cell biology.
8. In the methods section authors mention that they utilized both the male and female mice for this investigation. Was their any sex-specific effects? Which experiments were with male and which with female? They need to be clearly stating it.
Author Response
Major concerns:
1. The authors used ELOVL2-/- mice, and analyzed PBMCs and spleen to isolate and characterize T cells. They also need to investigate thymus, to overrule any developmental defects with respect to the T cells.
R: We thank the Reviewer for the suggestion but to obtain the thymus and analyze the % or activation of T cells is impossible because this Elovl2 KO model is no longer available in our laboratory (we were the only laboratory to have this specific model) and to do it again it would take us 1 year to start again from the embryos and to obtain the colony, which is one of the most difficult to obtain because it takes 5 generations to get a fully -/- mice, since males are sterile and female become sterile after 3 months.
However, although it might be possible that potential defects might exist in other tissues (thymus or even bone marrow), in this study we were only interested in peripheral blood and spleen. As soon as we will have the animal colony again, we will surely investigate other organs for possible mechanistic defects. In this study we only wanted to report for the first-time peripheral differences in T cell activation to set the basis for further studies. I hope the reviewer can accept this.
2-3. In Figures 2B and 2C, authors provide heatmaps. But, do not mention what they measure RNA or protein? There is no scale either and just visual quantification does not do justice. It will be of great help if authors show in main figures the RNA levels of the main cytokines which they study.
R: We want to point out that these heatmaps do not show RNA, but it’s protein because they are the very same data obtained by flow cytometry with intracellular antibodies against cytokines. The scale is shown on top of each heatmap and represent the % of cytokine expression in each cell subset. We provided to add this explanation in the figure legend.
4. The authors studied Th1 and Th17. Is there any effect of DHA on the regulatory T cells? The authors need to show and discuss this.
R: We appreciate this suggestion, but unfortunately not enough cells were available to also analyze the Treg subset and as explained in the first comment, we don’t have this animal model again inhouse and it would take us another year to do it. In our study we focused on the two most pathogenic cells and we added the potential role of Tregs in the discussion as a limit of this study.
5. IL-6 is a critical driver of Th17 cells. What happens to IL6 in absence of DHAs?
R: IL-6 is indeed significantly increased in absence of DHA and DHA-rich diet reduced this cytokine again and we have it already demonstrated it in our 2017 paper.
6. Authors need to show the status (expression or activity) of the critical transcription factors which are known to drive Th1, Th17 production and function.
R: As already stated in the discussion, here we lack the signaling mechanism (receptors and transcription factors) to do them, we would require to start from purify naïve CD T cells and to perform the classical 6-days polarization protocol during which Th1 and Th17 cells start to express their specific transcription factors at day 3-5. Here we didn’t start with naïve T cells to induce Th1 and Th17 but we analyze the already committed and differentiated Th1 and Th17 cells that circulate in the blood. To study the transcription factors, we would need to start from the naïve precursor cell and it would be a different rational to this study. Our aim was not to investigate the DHA role on the differentiation from naïve to Th1/Th17 cells but its role on the circulating and directly stimulated ones (with 3 different approaches: PMA/Ionomycin and anti-CD3/28 as well as with anti-CD3 in vivo) ones.
7. Authors correctly point out that the major tissue of DHA production is the Liver. It will be of immense significance if authors can genetically manipulate ELOVL2 levels solely in liver e.g. adenoviral-based methods and demonstrate whether this affects T cell biology.
R: Although it would be very interesting, we believe that the results would be redundant because the only DHA source that T cells use is the one from the liver. This is because DHA, besides the liver, is exclusively synthesized by testis, retina and the CNS, where DHA is fundamental for their functions (without DHA sperm cells cannot move, vision and cognition are severely impaired) and they certainly do not distribute DHA in the rest of the organs, which, instead would receive DHA only from the liver, which is in fact the organ that distribute every metabolite.
8. In the methods section authors mention that they utilized both the male and female mice for this investigation. Was their any sex-specific effects? Which experiments were with male and which with female? They need to be clearly stating it.
R: We thank the Reviewer for this suggestion. We added in the methods that both male and female mice were used. No sex-specific effect was observed in T cell responses and this was added in the text too.
Round 2
Reviewer 3 Report
The present revised MS is suitable for publication.